



# Efficiently modelling urban heat storage: an interface conduction scheme in the aTEB urban land surface model

Mathew J. Lipson[1], Melissa A. Hart[1], Marcus Thatcher[2]

[1]Climate Change Research Centre, UNSW and ARC Centre of Excellence for Climate System Science
[2]CSIRO Marine and Atmospheric Research, Aspendale, Australia

*Correspondence to*: M. Lipson (m.lipson@unsw.edu.au)

**Abstract.** Intercomparison studies of models simulating the partitioning of energy over urban land surfaces have shown the heat storage term is often poorly represented. In this study, two implicit discrete schemes representing heat conduction through urban materials are compared. We show that a well-established method of representing conduction systematically
underestimates the magnitude of heat storage compared with exact solutions of one-dimensional heat transfer. We propose an alternative method of similar complexity that is better able to match exact solutions at typically employed resolutions. The proposed interface conduction scheme is implemented in an urban land surface model and its impact assessed over a 15-month observation period for a site in Melbourne, Australia, resulting in improved overall model performance for a variety material parameter choices and aerodynamic heat transfer parameterisations. The proposed scheme has the potential to
benefit land surface models where computational constraints require a high level of discretisation in time and space, for example at neighbourhood/city scales, and where realistic material properties are preferred, for example in studies investigating impacts of urban planning changes.

## 1 Introduction

Urban structures change the climate of cities by affecting the partitioning of energy from natural and anthropogenic sources
(Oke, 1982). As climate affects human health, comfort and the energy use of buildings, many land surface models have been developed to simulate the impacts of different urban forms (Grimmond et al., 2009). An important term in the energy balance of atmosphere-surface interactions is storage heat flux density ($Q_S$): the net flow of heat per unit area into and out of materials. In highly urbanised areas, $Q_S$ becomes the dominant term of the daytime urban energy balance (Cleugh and Grimmond, 2012). At night, excess heat absorbed by dense urban materials is released back into the atmosphere. The altered
behaviour of $Q_S$ fundamentally affects environmental processes such as atmospheric stability, the evolution of the boundary layer, convection, and pollution dispersion, and is key in establishing urban heat islands (Barlow, 2014). However, the most recent urban model intercomparison project found most participants significantly under-represented heat storage, with average midday $Q_S$ bias errors of -50 W m$^{-2}$ (Best and Grimmond, 2014a). In a world of rapid urbanisation and changing global climate, researchers and planners are interested in efficient and accurate models that are able to investigate urban





climate impacts. Improving heat storage representations at the neighbourhood to city-scale will benefit those studies. This paper proposes an alternative method to represent heat storage in urban land surface models. Although the proposed scheme is similar in complexity and design to a current well-established method, the analysis undertaken here indicates the alternate method is more accurate at relevant resolutions and reduces overall energy flux errors of an urban land surface model.

Further in this introduction, key issues in measuring and modelling urban heat storage are discussed. Section 2 describes the evaluated conduction schemes. Section 3 analyses their performance in isolation in an idealised environment. Section 4 assesses their impact when implemented in an urban land surface model. Section 5 discusses and concludes findings.

### 1.1 Measuring storage heat flux in cities

In simple environments it is possible to measure $Q_S$ directly through a network of heat flux plates (Nunez and Oke, 1977),
but the variety of urban materials, orientations, sky-view factors and internal building environments in a typical city make direct measurement at desired scales impractical. In order to practically account for the full diversity of surfaces in a heterogeneous urban landscape, the most commonly accepted method to measure $Q_S$ at the neighbourhood scale is by calculating heat storage as a residual of the urban energy balance (Roberts et al., 2006). Following Oke (1988) a full representation of the urban energy balance (in W m$^{-2}$) is:

$$Q^* + Q_F = Q_H + Q_E + Q_S + Q_A, \tag{1}$$

where inputs of net all-wave radiation flux ($Q^*$) and anthropogenic heat flux ($Q_F$) are balanced by sensible and latent turbulent heat fluxes ($Q_H$ and $Q_E$), storage ($Q_S$), and net advection flux ($Q_A$). Using radiometers and eddy covariance techniques, $Q^*$, $Q_H$ and $Q_E$ can be measured directly, and are considered representative of the neighbourhood if instruments are at a sufficient height above ground where effects of individual roughness elements are blended (Cleugh and Oke, 1986).
$Q_A$ is often considered negligible and excluded if areas adjoining observation sites have similar urban characteristics (Roberts et al., 2006). $Q_F$ can be estimated through energy use and population density analysis (e.g. Sailor and Lu, 2004). Then, rearranging Eq. (1) residual ($\Delta$) heat storage is:

$$\Delta Q_S = (Q^* + Q_F) - (Q_H + Q_E). \tag{2}$$

The residual approach has the inherent problem of accumulating all observational errors, which can be significant for
turbulent fluxes (Wilson et al., 2002). Estimates of $Q_F$ and assumptions regarding $Q_A$ add to overall uncertainty. The daytime errors in $\Delta Q_S$ observations used in this study have previously been estimated at 23%, doubling at night (Best and Grimmond, 2014b). Given the significant observational uncertainties, this study first evaluates conduction schemes with exact solutions to one-dimensional heat transfer, before comparing with observations of $\Delta Q_S$ calculated as a residual of the urban energy balance.



### 1.2 Simulating storage heat flux in cities

At scales of individual buildings, $Q_S$ of walls and roofs can be accurately modelled by solving the three-dimensional heat conduction equation at high resolution. However, simulating urban weather and climate over larger scales requires simplification in order to keep computation practical (Martilli, 2007). Some urban models replace heat transfer calculations

with empirical functions of net radiation (e.g. Grimmond and Oke, 2002), or with idealised representations like the force-restore method (e.g. Porson et al., 2010). Another common method, used by 23 of 32 participants of the First International Urban Land Surface Model Comparison Project (PILPS-Urban), is to calculate a discretised version of the heat conduction equation (Grimmond et al., 2010). Best and Grimmond (2014b) found models using this method were better at simulating $Q_S$, although all categories of models predicted $Q_S$ outside observational error in more than 50% of daytime intervals.

Overall, models partitioned too little incoming energy into storage and too much into other fluxes, a result supported by a previous intercomparison at a different site (Best and Grimmond, 2015; Grimmond et al., 2010). Although other studies evaluating individual urban models have found good agreement between simulated and observed $Q_S$ (e.g. Roberts et al., 2006), PILPS-Urban showed that most models have difficulty predicting $Q_S$ at sites that had not previously been used to evaluate a model, where no parameter optimisation had been undertaken. This study suggests a method to improve $Q_S$

prediction using an urban model that is conceptually similar to many participants of PILPS-Urban.

### 1.3 Material thermal parameters

A potential weakness in evaluating urban model performance is the wide variety of parameters that could describe urban materials. Urban models are sensitive to material thermal parameter variation, particularly the magnitude and phase of storage heat flux and sensible heat flux (Oleson et al., 2008b). Selecting the best thermal parameters is not necessarily

straightforward; as land surface models are a simplified representation of reality, model material parameters should only be viewed as abstract representations of observed physical quantities (Gupta et al., 1999). Researching and inputting realistic values based on local material parameters requires considerable effort, and can result in little improvement to performance, or even degrade it (Loridan and Grimmond, 2012). In PILPS-Urban, average model performance in $Q_S$ worsened as more site-specific (realistic) material thermal properties were provided to participants (Best and Grimmond, 2014b). As such,

modellers sometimes use material parameters that do not match with the realities of the simulation site, but produce results that match well with observations (i.e. optimised parameters). This may pose problems for studies at new sites with different conditions, and studies that wish to ascertain impacts of changing urban materials. There is therefore interest in improving $Q_S$ performance using more realistic material parameters. Additionally, some models use a single set of material parameters to describe homogenous urban surfaces, while others allow distinct layers with different parameters composing a composite

assembly. In order to cover a range of modelling possibilities, we evaluate optimised, realistic, homogenous and layered material parameters by drawing from five material dataset sources, described below and in Appendix A.





**CLMu (realistic, layered):** From a dataset of global urban characteristics (Jackson et al., 2010) intended for use in the Community Land Model – Urban (CLMU) (Oleson et al., 2008a), or other global climate models. The database catalogues the properties of 32 common walls and roofs from around the world. Attempts are made by Jackson et al. to reconcile the whole wall/roof thermal conductivity with real world values by estimating the effects of thermal bridging, air leakage, and

poor construction. For our analysis, all 10-layer composite walls and roofs were collapsed through each iteration down to two layers using depth-weighted averages of conductivity and heat capacity, resulting in 288 representations of walls and roofs of varying thermal characteristics and complexity.

**SITE (realistic, layered):** As presented in PILPS-Urban Phase 2 (Grimmond et al., 2011). Characteristics were derived from an area and depth weighted average of material thermal properties at the observation site. Roofs and walls were an aggregate

of metal, terracotta, concrete and asbestos, insulation, lightweight framing and plasterboard, separated into four layers: external skin, structure, insulation and internal lining.

**WRF (optimised, homogenous):** From the WRF/urban integrated urban modelling system v3.2 (Chen et al., 2011), which includes the SLUCM and BEP urban schemes. In WRF, three default sets of parameters are available for various densities of urban land cover, here we use the low intensity residential set following the observation site's classification by Loridan and

Grimmond (2012). The WRF default parameters represent a generic homogenous material with a heat capacity and conductivity similar to lightweight concrete throughout.

**UZE (optimised, homogenous):** From updated WRF/urban parameters described in Loridan and Grimmond (2012) based on results of a multi-objective optimisation algorithm to minimise root mean square error (RMSE) over 15 urban locations. New parameter values were recommended for three categories of urban areas based on Urban Zone for Energy exchange

(UZE: Loridan and Grimmond, 2011) and were subsequently included in releases of WRF/urban as an optional dataset. We use the medium urban category following the observation site's classification by Loridan and Grimmond (2012).

**aTEB (optimised, layered):** From the ECOCLIMAP database (Masson et al., 2003) on which TEB (Masson, 2000) defaults are based, but with increased layer depths per Thatcher and Hurley (2012). The walls and roofs are not representative of typical building methods in Australia, but nonetheless give reasonable results for generic Australian cities. The roof is a

layered composite with thermal conductivity and heat capacity of dense concrete, aerated concrete and insulation; the walls a composite of concrete and insulation and the road/soil of asphalt and dry soil.

## 2 Description of conduction representations

Three calculation methods of heat storage are compared: two discrete schemes and an exact solution. The two discrete schemes lump a material's heat capacitance at a temperature node and calculate solutions numerically at each timestep. The

exact method calculates continuous harmonic solutions to a periodic forcing. All three solve Fourier's law:

$$q = \lambda \frac{\partial T}{\partial d}, \tag{3}$$

and the one-dimensional continuity equation:





$$C\frac{\partial T}{\partial t} = \frac{\partial q}{\partial d}, \tag{4}$$

where $q$ is the conduction heat flux density [W m$^{-2}$], $C$ is volumetric heat capacity [J m$^{-3}$ K$^{-1}$], $\lambda$ the thermal conductivity [W m$^{-1}$ K$^{-1}$], $T$ the temperature [K], $d$ the depth [m] and $t$ is time [s].

### 2.1 Half-layer scheme

5    A common discretised approach is to locate the temperature node centrally within a homogenous layer – a half-layer scheme (Fig. 1 (a)). This approach is used in many urban land surface schemes, for example Town Energy Budget (TEB) (Masson, 2000), Single-Layer Urban Canopy Model (SLUCM) (Kusaka et al., 2001), Building Effect Parameterization (BEP) (Martilli et al., 2002), Community Land Model – Urban (CLMU) (Oleson et al., 2008a), Vegetated Urban Canopy Model (VUCM) (Lee and Park, 2008), and the Australian Town Energy Budget (aTEB) (Thatcher and Hurley, 2012). Models utilising this

10    method vary in their spatial and temporal resolution, but typically resolve between 1 and 10 substrate temperature nodes, at between 5 and 60 minute timesteps (Grimmond et al., 2009, 2010). The half-layer method is based on well-established land surface models representation of thermal conduction through soil (e.g. Oleson et al., 2010), and is also used in multi-layer snow and sea ice models (e.g. West et al., 2016).

A discretised form of the conduction equation (Eq. 3) is:

$$q_{k,k+1}^m = \frac{\lambda_{k,k+1}}{d_{k,k+1}}(T_k^m - T_{k+1}^m), \tag{5}$$

where $T_k^m$ is the temperature at the $k$th node at timestep index $m$. Conduction between temperature nodes occurs through material layers with homogenous and invariant thermal characteristics. Where two adjacent layers $i$ and $i+1$ have different depths $d$ or conductivities $\lambda$, then the half-layer scheme will represent an effective conductance [W m$^{-2}$ K$^{-1}$] between temperature nodes $k$ and $k+1$ as:

$$\frac{\lambda_{k,k+1}}{d_{k,k+1}} = \frac{1}{\frac{1}{2}\left(\frac{d_i}{\lambda_i} + \frac{d_{i+1}}{\lambda_{i+1}}\right)} = \frac{1}{\frac{1}{2}(R_i + R_{i+1})}, \tag{6}$$

where resistance $R_i = d_i/\lambda_i$ (with $i = k$ hereafter, $i$ is dropped). Then, combining Eq. (5) and (6) with the discretised form of the continuity Eq. (4) over timestep length $\Delta t$, the general implicit formulation of the half layer scheme is:

$$C_k d_k \left(\frac{T_k^{m+1} - T_k^m}{\Delta t}\right) = q_{k-1,k}^{m+1} - q_{k,k+1}^{m+1} = \frac{T_{k-1}^{m+1} - T_k^{m+1}}{\frac{1}{2}(R_{k-1} + R_k)} - \frac{T_k^{m+1} - T_{k+1}^{m+1}}{\frac{1}{2}(R_k + R_{k+1})}, \tag{7}$$

Heat storage flux density through to the next timestep ($Q_S^{m+1}$) is the sum of the change in energy stored in individual layers:

$$Q_{S,half-layer}^{m+1} = \sum_{k=1}^{n} \frac{C_k d_k (T_k^{m+1} - T_k^m)}{\Delta t}. \tag{8}$$

Energy conservation is demonstrated by the equivalency of Eq. (7) and Eq. (8) for layers 1 to $n$, as inner conduction fluxes ($q_{1,2}^{m+1} \cdots q_{n-1,n}^{m+1}$) cancel, leaving heat storage flux density equal to outer conduction terms:

$$Q_{S,half-layer}^m = q_{ext}^m - q_{int}^m = \frac{T_{ext}^m - T_1^m}{\frac{1}{2}R_1 + R_{ext}} - \frac{T_n^m - T_{int}^m}{\frac{1}{2}R_n + R_{int}}, \tag{9}$$





where $q_{ext}^m$ is the external admittance flux which drives transience in the system, $q_{int}^m$ is the transmittance flux into the building, $T_{ext}^m / T_{int}^m$ are external/ internal environmental temperatures, and $R_{ext} / R_{ext}$ are external/ internal surface thermal resistances.

## 2.2 Interface scheme

While the half-layer scheme lumps capacitance at the centre of layers, an alternative approach is to lump capacitance at the interface between layers (Figure 1(b)). Since the paths of conduction between nodes are now completely within homogenous layers, Eq. (6) simplifies to:

$$\frac{\lambda_{k,k+1}}{d_{k,k+1}} = \frac{\lambda_i}{d_i} = \frac{1}{R_i},\tag{10}$$

where $i = k$. The capacitance of the temperature node now takes half the value of the adjacent layers; in effect we have
swapped the need for an effective conductance for an effective capacitance. The general implicit formulation for the interface scheme is:

$$\left(\frac{C_{k-1}d_{k-1}+C_k d_k}{2}\right)\left(\frac{T_k^{m+1}-T_k^m}{\Delta t}\right) = q_{k-1,k}^{m+1} - q_{k,k+1}^{m+1} = \frac{T_{k-1}^{m+1}-T_k^{m+1}}{R_{k-1}} - \frac{T_k^{m+1}-T_{k+1}^{m+1}}{R_k},\tag{11}$$

with outer temperature nodes represented by half the outer layer heat capacity only, i.e. at $k = 1$; $C_0 d_0 = 0$ and $k = n + 1$; $C_{n+1}d_{n+1} = 0$ for an $n$ layer system.

Total heat storage flux density for the interface scheme is:

$$Q_{S,interface}^{m+1} = \frac{1}{2}\frac{C_1 d_1(T_1^{m+1}-T_1^m)}{\Delta t} + \sum_{k=2}^{n}\frac{1}{2}\frac{(C_{k-1}d_{k-1}+C_k d_k)(T_k^{m+1}-T_k^m)}{\Delta t} + \frac{1}{2}\frac{C_n d_n(T_{n+1}^{m+1}-T_{n+1}^m)}{\Delta t}.\tag{12}$$

Again, energy conservation is demonstrated through the equivalency of Eq. (11) and Eq. (12), which after cancelling inner conduction terms leaves heat storage flux density equal to outer conduction terms:

$$Q_{S,interface}^m = q_{ext}^m - q_{int}^m = \frac{T_{ext}^m-T_1^m}{R_{ext}} - \frac{T_{n+1}^m-T_{int}^m}{R_{int}}.\tag{13}$$

For the half-layer and interface conduction schemes, net heat storage calculated as a sum of the change of energy in each layer is equal to the sum of fluxes at external and internal surfaces. Although energy is conserved within each scheme, Eq. (9) and (13) show the calculated net heat storage fluxes are different.

## 2.3 Exact solution

Here we use the admittance procedure (Butcher, 2006; Davies, 1973), which calculates exact solutions to planar heat transfer
through a series of homogenous layers when subject to a steady sinusoidal forcing on one side, with a fixed temperature on the other. The international standard ISO 13786:2007 documents the method. The exact solution to the heat storage flux density $Q_{S,exact}$ for a composite wall is the admittance flux density minus the transmittance flux density:

$$Q_{S,exact}(t) = q_{ext}(t) - q_{int}(t) = \left|\frac{1-H_{22}}{H_{12}}\right| \sin\left(\frac{2\pi}{P}t + \arg\left(\frac{1-H_{22}}{H_{12}}\right)\right),\tag{14}$$





where $P$ is the period of sinusoidal forcing and $H_{12}$ and $H_{22}$ are components of a 2×2 complex-valued heat transfer matrix calculated via multiplication of individual heat transfer matrices over $n$ homogenous layers:

$$\begin{pmatrix} H_{11} & H_{12} \\ H_{21} & H_{22} \end{pmatrix} = \begin{pmatrix} 1 & -R_{ext} \\ 0 & 1 \end{pmatrix} \cdot \begin{pmatrix} Z_{11}^1 & Z_{12}^1 \\ Z_{21}^1 & Z_{22}^1 \end{pmatrix} \cdot \begin{pmatrix} Z_{11}^2 & Z_{12}^2 \\ Z_{21}^2 & Z_{22}^2 \end{pmatrix} \cdots \begin{pmatrix} Z_{11}^{n-1} & Z_{12}^{n-1} \\ Z_{21}^{n-1} & Z_{22}^{n-1} \end{pmatrix} \cdot \begin{pmatrix} Z_{11}^n & Z_{12}^n \\ Z_{21}^n & Z_{22}^n \end{pmatrix} \cdot \begin{pmatrix} 1 & -R_{int} \\ 0 & 1 \end{pmatrix} \quad (15)$$

where $R_{ext}$ and $R_{int}$ are external and internal surface thermal resistances and $Z_{ab}^i$ a component of the $i$th layer heat transfer matrix, from outside in. Components of the $i$th heat transfer matrix are:

$$Z_{11}^i = Z_{22}^i = \cosh\frac{d_i}{\delta_i}\cos\frac{d_i}{\delta_i} + j\sinh\frac{d_i}{\delta_i}\sin\frac{d_i}{\delta_i}, , \quad (16)$$

$$Z_{12}^i = -\frac{\delta_i}{2\lambda_i}\left[\sinh\frac{d_i}{\delta_i}\cos\frac{d_i}{\delta_i} + \cosh\frac{d_i}{\delta_i}\sin\frac{d_i}{\delta_i} + j\left(\cosh\frac{d_i}{\delta_i}\sin\frac{d_i}{\delta_i} - \sinh\frac{d_i}{\delta_i}\cos\frac{d_i}{\delta_i}\right)\right], \quad (17)$$

$$Z_{21}^i = -\frac{\lambda_i}{\delta_i}\left[\sinh\frac{d_i}{\delta_i}\cos\frac{d_i}{\delta_i} - \cosh\frac{d_i}{\delta_i}\sin\frac{d_i}{\delta_i} + j\left(\sinh\frac{d_i}{\delta_i}\cos\frac{d_i}{\delta_i} + \cosh\frac{d_i}{\delta_i}\sin\frac{d_i}{\delta_i}\right)\right], \quad (18)$$

where $j$ is the imaginary unit and:

$$\delta_i = \sqrt{\frac{\lambda_i P}{\pi c_i}}, \quad (19)$$

is the periodic penetration depth of the $i$th layer.

## 3 Idealised evaluation

The half-layer and interface discrete schemes are compared with exact solutions of heat transfer with a sinusoidal temperature forcing. Performance statistics are based those used in PILPS-Urban Phase 2, as described in Phase 1 (Grimmond et al., 2010).

### 3.1 Idealised method

The external boundary is forced by $T_{ext}$, a sinusoidal temperature variation of one degree over a 24-hour period representing the combined effects of external environment temperature and incident radiation (or sol-air temperature):

$$T_{ext} = T_0 + \sin\left(\frac{2\pi}{P}t\right), \quad (20)$$

where $T_0 = 290$ K is the average external temperature. In this study, the internal boundary environment $T_{int}$ is fixed, equal to $T_0$. Boundaries have fixed surface thermal resistances representing both convective and radiative transfer. Values are standard horizontal heat transfer rates set out in ISO6946, where $R_{ext} = 0.04$ and $R_{int} = 0.13$ m$^2$ K W$^{-1}$ (ISO, 2007).

For both discrete schemes, a linear system of equations describing the temperature evolution of each node is generated and solved by decomposition and back-substitution of a tridiagonal matrix (Thomas' algorithm). Discrete schemes are run for 6 forcing periods of 24 hours, with the first five periods discarded as spin-up, and the last compared with the exact solution. For the exact solution, components of the heat transfer matrix are calculated numerically and harmonic solutions computed.



### 3.2 Idealised results

#### 3.2.1 Idealised results: high-resolution simulation

Discrete schemes are tested at very high resolution (200 layers, each 1 mm deep) in order to test code validity. Table 1 shows the normalised standard deviation (SD) of high spatial resolution simulations over various timesteps, where an SD of 1.0

means amplitudes of discrete and exact solutions match. At the higher time and space resolutions, both schemes converge towards the exact solution. As the temporal resolution is decreased to 30-minute timesteps amplitudes of $Q_S$ reduce in both schemes.

#### 3.2.2 Idealised results: realistic material parameters

We now test the performances of the discrete schemes at space and time resolutions more practical and typical of urban land

surface models. The twenty wall and twelve roof configurations described in the CLMu database were tested at 30-minute timesteps over various levels of complexity. In Fig. 2, errors for 2–10 layer configurations are plotted individually (288 total), along with layer means. Of the two discrete schemes, the interface scheme has an average normalised SD closer to the exact solution for each layer up to ten (Fig. 2(a)). The mean normalised SD of $Q_S$ for the interface scheme begins above the exact solution at two layers, but decreases at as resolution increases. For the half-layer scheme, SD begins below the exact

solution and increases as resolution increases. For the interface scheme at 30-minute timesteps, mean spatial and temporal discretisation normalised SD errors almost cancel at four layers, but never cancel for the half-layer scheme.

Figure 2 (b) shows the layer mean of the interface scheme mean absolute errors (MAE) are smaller than the half-layer scheme. The normalised MAE for the half-layer scheme monotonically decreases as the number of layers increases. Time and space discretisations both lead to negative bias in $Q_S$, so MAE is minimised by increasing the number of layers

indefinitely. On the other hand, the interface scheme average MAE declines until a minimum at four layers, where the positive bias of spatial discretisation approximately balances the negative bias of temporal discretisation. Using the same method for 1, 15, 30 and 60-minute timesteps, we find the optimal numbers of layers for the interface scheme are: 8, 6, 4 and 2, while the half-layer scheme only improves at higher resolutions (Appendix C). At higher resolutions, the interface scheme provides less benefit over the half-layer scheme, but is still able to simulate $Q_S$ more accurately for a range of realistic wall

and roof assemblies.

Having evaluated the performance of the two schemes over increasing material complexity, we evaluate performance for different composite thermal characteristics. Figure 3 shows the normalised SD and the MAE for each scheme at 30-minute timesteps for the 288 CLMU walls and roofs, plotted against the cyclic heat capacity of the composite material (calculated per ISO 13786:2007). Cyclic heat capacity is similar to thermal admittance (or inertia or effusivity) in that it is a product of

volumetric heat capacity and conductivity, but it also accounts for a material's composite nature, periodic penetration depth and heat lost through transmittance. Materials with low cyclic heat capacity (e.g. lightweight framed walls or sheet metal) do not absorb and store as much heat over the diurnal cycle as do materials with high cyclic heat capacity (e.g. concrete or



brick). In contrast to discrete layers, cyclic heat capacity varies continuously, so the response of each scheme is represented as a locally weighted linear regression (LOWESS) (Cleveland, 1979). The thicker lines are representative of all 288 material configurations, while the thin dashed lines represent a subset of walls and roofs with four layers (the number of layers used later in the urban model analyses).

Figure 3 (a) shows the interface scheme LOWESS of normalised SD is closer to the exact solution for all values of cyclic heat capacity represented in the CLMu database. The half-layer scheme's LOWESS of normalised SD shows an increasing negative bias for larger cyclic heat capacity, while the interface scheme LOWESS is less steep. Figure 3 (b) plots MAE (non-normalised) and shows schemes have greater difference in absolute errors for assemblies with higher cyclic heat capacity. Heat storage becomes a larger proportion of the energy balance in neighbourhoods with more heat capacity, so a scheme that

is better able to represent $Q_S$ in those instances is beneficial.

### 3.2.3 Idealised results: typical material parameters

We now evaluate the performance of the discrete schemes using wall and roof characteristics that are typical of previous urban modelling studies. Figure 4 displays the results of four walls, one realistic (SITE) and three optimised (WRFu, UZEm and aTEB). All walls are made up of four layers and later analysed in the aTEB urban model (Sect 4.2). The upper panels

show $Q_S$ amplitude through a 24-hour forcing period, while the lower panels show the error from the exact solution at each 30-minute timestep. In each case, the half-layer scheme under-simulates the diurnal amplitude of $Q_S$, while the interface scheme matches more closely with the exact solution. The mean absolute error (MAE) normalised by the absolute mean of the exact solution (Fig. 4 lower panel) is smaller for walls represented with the interface scheme.

Fig. 5 displays results for four-layered roofs, which do not show the same degree of improvement from the interface scheme

as the walls. Errors for both schemes are more pronounced in the UZE and aTEB roofs, which have greater total depths and therefore higher spatial discretisation errors (roof and wall characteristics listed in Appendix A). Overall the interface scheme improves performance of these four-layer representations.

As a consequence of the method of discretisation, the interface scheme has one additional temperature node over the half-layer scheme for any given number of layers (see Fig. 1). In Fig. 2, the interface average errors are lower than half-layer

average errors for assemblies with $n + 1$ layers, implying the benefit of the interface scheme is broader than simply adding an additional node. A further test can be performed for the homogenous assemblies in Fig 4 and 5 (WRF and UZE walls/roofs) giving the half-layer scheme an extra node by dividing the first layer equally; then both schemes are represented by the same number of nodes, and the first node has the same heat capacity. When MAE is calculated no five-layered half-layer system outperforms the corresponding four-layered interface system.



## 4 Observational evaluation

The discrete schemes are now compared within an urban land surface model (aTEB) forced by observations using the methodology of the PILPS-Urban Phase 2 (Grimmond et al., 2011) in order to assess overall impact on flux predictions. The same four wall and roof thermal parameters from Sect. 3.2.3 were input into aTEB while all other parameters including

urban morphology, roof/wall radiative properties were kept constant, including road/soil properties to minimise differences in hydrology (refer Appendix A for values).

### 4.1 Observational methods

#### 4.1.1 Description of urban model: aTEB

The proposed interface scheme was implemented in the Australian Town Energy Budget (aTEB) urban land surface model

(Thatcher and Hurley, 2012). aTEB was developed to act as the urban component of a regional or global climate model, so takes the highly efficient building-averaged approach where the generic urban unit is an infinite street canyon (Nunez and Oke, 1977). Canyon surfaces (walls, road, snow and vegetation) are connected to a bulk canyon air layer via an aerodynamic resistance network. Roofs and canyon air are then connected in parallel to the overlying atmosphere.

Although written from the ground up, aTEB is conceptually based on the influential Town Energy Budget (TEB) urban

canopy model (Masson, 2000) with some modifications for Australian conditions. Modifications include:

- In-canyon vegetation for suburban areas represented by a big-leaf model, adapted from Kowalczyk et al. (1994) but with a largely reduced set of prognostic variables.
- Air-conditioning component which pumps waste heat into canyons and prevents buildings acting as energy sinks during high temperature periods, allowing energy closure at each timestep.

- Two-wall canyon allowing radiative interactions between a sunlit and shaded walls, and a canyon airflow paramaterisation with venting and recirculating regions, each integrated through 180º for all possible street orientations, adapted from Harman et al. (2004a, 2004b).

aTEB would be categorised as a 'complex' urban model following the methodology of Grimmond et al. (2010, 2011), primarily because of its canyon based approach. Conceptually similar models include TEB (Masson, 2000), SLUCM

(Kusaka et al., 2001) and CLMU (Oleson et al., 2008a). A significant differentiator amongst conceptually similar models is the parameterisation of heat exchange between canyon surfaces and turbulent air. By default, the SLUCM uses a form of the Jürges formula (1924), while TEB and CLMU use a form of Rowley et al. (1930). An alternative approach was developed by Harman et al. (2004b) where aerodynamic conductance is separately calculated for each canyon surface based on the airflow in different regions of the canyon for different canyon geometries. Urban canopy models that utilise forms of the Harman

circulation scheme include the Single Column Reading Urban Model (SCRUM) (Harman and Belcher, 2006), the Met Office Reading Urban Surface Exchange Scheme (MORUSES) (Porson et al., 2010) and aTEB (Thatcher and Hurley, 2012). In order to assess how the proposed conduction scheme is affected by different aerodynamic heat transfer parameterisations, the





Jürges, Rowley and Harman methods have been implemented in aTEB as described in Appendix B. Further details on aTEB is available in Thatcher and Hurley (2012) and Luhar et al. (2014).

**4.1.2 Observational data**

Observational data were obtained from flux tower measurements in a suburban site in Melbourne, Australia (Coutts et al.,
2007a, 2007b). Data includes up-welling and down-welling long and shortwave radiation ($K_\uparrow$, $K_\downarrow$, $L_\uparrow$, $L_\downarrow$), net radiation ($Q^*$), turbulent heat fluxes ($Q_H$, $Q_E$), air temperature, pressure, wind, humidity and rainfall. Sampling rates were between 1 and 10 Hertz, block averaged to 30-minute intervals over 474.4 continuous days from 13 August 2003 to 28 November 2004. Measurements were taken at 40 m above ground, at a height where the effects of individual buildings were sufficiently blended so that measurements were considered representative of the neighbourhood.

The gap-filled data used here is identical to that used in the First International Urban Land Surface Model Comparison Project (PILPS-Urban) Phase 2 (Grimmond et al., 2011), from which our evaluation methodology follows, that is:

- Observed downwelling radiation, air temperature, pressure, wind, humidity and rainfall data were used as forcing data to run the urban model offline, i.e. without the need to be coupled to a larger atmospheric/ earth system model.
- Observed upwelling radiation, turbulent heat and residual heat storage flux observations were compared with model
output to evaluate the performance of the model.
- The initial 108.4 days of observation were treated as spin-up and excluded from analysis.
- The remaining 366 days were analysed, but if any flux were missing in a time interval, all data in that interval were ignored, resulting in 8520 usable half-hour time intervals.

The site at Preston, Melbourne is typical of low to medium density suburban housing in Australia, with detached one to two
story brick, timber and steel framed buildings, separated by roads, lawn and large trees. The site is classified in Best and Grimmond (2014b) as a local climate zone (LCZ) 6 (Stewart and Oke, 2012) or as an Urban Zone for Energy exchange (UZE) medium density (Loridan and Grimmond, 2011, 2012).

**4.2 Observational results**

**4.2.1 Observational results: heat storage**

Figure 6 compares the residual storage heat flux density ($\Delta Q_S$) of observations and the urban model by calculating a mean flux for each hour of the day over the 12-month evaluation period. The method of determining observed $\Delta Q_S$ as the residual of the surface energy balance includes inherent uncertainty (refer Sect. 1.2), however the relatively long observation period used here reduces stochastic error. The upper panels show the mean hourly flux density over the diurnal cycle, the lower panels show error from observations. For each of the four datasets, the amplitude of the diurnal storage heat flux density
increases when using the interface scheme compared to the half-layer scheme, similar to the behaviour seen in the exact analysis. As the half-layer method generally under-represents the magnitude of $\Delta Q_S$, the interface scheme is better able to



represent the observed storage heat flux over the full diurnal cycle. The interface scheme also improves simulated storage heat flux at the 25th and 75th quartiles, particularly during the day.

Half-hourly $Q_S$ performance statistics of aTEB using the Harman aerodynamic heat transfer method are listed in Table 2. For the four material databases, the interface scheme improves error metrics in almost every case. These statistics are comparable
to those reported in the PILPS-Urban Phase 2 intercomparison project, which used the same observational data (Grimmond et al., 2011). In that study, heat storage was the most poorly represented energy flux (Best and Grimmond, 2015), where the best $\Delta Q_S$ RMSE of any model in the final stage was 53 W m$^{-2}$, and the mean of participating models was 65 W m$^{-2}$.

The interface scheme in general improves the urban simulation performance in $\Delta Q_S$ for these four datasets. However, improving the performance in $\Delta Q_S$ may simultaneously deteriorate performance in another flux of the energy balance
(Loridan et al., 2010; Loridan and Grimmond, 2012), and so assessing the impact of the proposed conduction scheme on other energetic fluxes is necessary.

### 4.2.1 Observational results: other fluxes

In Fig. 7, a Taylor diagram (Taylor, 2001) extends the statistical evaluation of the two conductions schemes to include sensible and latent heat, and upwelling longwave radiation fluxes. Shortwave radiation flux is omitted because it is
unaffected by the conduction schemes, and downwelling radiation fluxes are prescribed. A Taylor diagram is useful in determining whether a change in the difference error (centred root mean square error: cRMSE) occurs from a change in variance (standard deviation: SD) or a change in pattern correlation (Pearson's correlation coefficient: $r$). cRMSE and SD are normalised by the standard deviation of corresponding flux observations to easily compare fluxes with different value ranges and variances. The two conductions schemes are tested with the four material datasets (Appendix A), along with three
different aerodynamic heat transfer methods (Appendix B) for a total of 24 simulations. We also plot the anonymous results of participants in the PILPS-urban Phase 2 intercomparison (Grimmond et al., 2011). Simulations using the half-layer scheme are represented by unfilled coloured markers and the interface scheme with filled markers, with the two conduction schemes connected by a line.

For residual storage heat flux ($\Delta Q_S$), the interface scheme improves cRMSE for all simulations. The reduced cRMSE is
primarily from improvement in normalised SD. Overall best performance is achieved with the aTEB default material parameters, followed by UZE, WRF and SITE. For each material dataset, the Harman heat transfer method performs better than the Rowley or Jürges methods. The length of the connecting line between conduction schemes compared to the tight grouping of the three heat transfer methods indicates the conduction scheme generally has greater impact on performance of $\Delta Q_S$ than the different aerodynamic heat transfer methods. The spread of the material datasets indicates the model is
sensitive to material thermal parameter choices.

For upwelling longwave radiation flux ($L_\uparrow$), the interface scheme improves cRMSE for all simulations. Improvements in both normalised SD and $r$ contribute to the lower cRMSE. Rowley and Harman methods perform better than the Jürges method, however the model is more sensitive to material dataset and the conduction scheme choice.




For sensible heat flux ($Q_H$), the interface scheme improves cRMSE for all simulations. The reduced cRMSE results primarily from improved normalised SD, as correlation is generally flat or slightly degraded. The Jürges and Harman methods perform better than the Rowley method. For $Q_H$, the difference between conduction schemes is less pronounced than for $\Delta Q_S$ and $L_\uparrow$.

For latent heat flux ($Q_E$), the impact of the interface scheme on cRMSE is mixed, but on average degrades performance. No clear pattern emerges from either SD or $r$, and all simulations are tightly grouped.

The mean change in performance from half-layer to interface schemes are presented in Table 3, with improvements in bold.

## 5 Discussion and Conclusion

We evaluated the performance of two implicit discrete schemes that represent heat conduction through urban surfaces. The half-layer scheme is well established and widely used in land surface models for urban structures, soil, snow and ice. It lumps heat capacitance at nodes centred within discrete layers. We proposed an alternative scheme of similar complexity that lumps heat capacitance at the interface between layers. We used two independent methods to evaluate the schemes: comparison with exact solutions to heat transfer in an idealised environment, and comparison with a long time-series observations for an urban site with heat storage calculated as a residual of the urban energy balance.

In the idealised evaluation (Section 3), a series of multi-layered assemblies of various complexities were subjected to a sinusoidal temperature forcing representing the diurnal cycle. The half-layer scheme was found to systematically under-estimate $Q_S$ magnitude compared to exact solutions of one-dimensional heat transfer, while the interface scheme better matched exact solutions. Time and space discretisation errors of the half-layer scheme were both negatively signed, leading to under-representation of $Q_S$. For the interface scheme, discretisation errors were oppositely signed, reducing average error. An optimal combination of space and time discretisation can be found to approximately cancel bias errors; for 30-minute timesteps, the interface optimum was four material layers. Overall, the interface scheme provided greatest benefit for simpler representations (fewer layers), and for materials with larger cyclic heat capacity (higher thermal mass).

In the observational evaluation (Section 4), we assessed the impact of implementing the interface scheme on performance of an urban land surface model (aTEB) that is conceptually similar to many participants of the PILPS-Urban intercomparison project. We evaluated four material parameter datasets and three common aerodynamic heat transfer parameterisations over a 15-month observation period for a site in Melbourne, Australia. In the urban model, the interface scheme tended to increase the diurnal magnitude of $Q_S$. In all simulations, heat storage was under-represented compared with observations, so increasing $Q_S$ magnitude improved performance. Each flux is linked via the urban surface energy balance and so all fluxes were affected by a change in heat storage. As more energy was partitioned into storage, less was available for the turbulent fluxes and urban skin surface temperatures were lower. As both $Q_H$ and $L_\uparrow$ were over-represented, the interface scheme improved performance in these fluxes too. For $Q_E$, the impact of the interface scheme was mixed, but on average degraded performance. Material thermal characteristics were not chosen to achieve the lowest flux errors, but to assess the impact of the alternative conduction scheme for typical simulation setups. Four previously published material datasets were evaluated:





SITE, WRF, UZE and aTEB. The interface scheme improved performance for all datasets, however overall errors were still large. The SITE material parameters, which were supposed to match most closely with realities of the site, performed the worst. The optimised material datasets with unrealistically large storage capacities performed better. Three common canyon surface aerodynamic heat transfer methods were also evaluated: Rowley, Jürges and Harman. The Harman method had lower

$Q_S$ errors than equivalent experiments using Rowley or Jürges, but for other fluxes there was no clear best method. Overall, the urban model was more sensitive to material thermal dataset and conduction method than to aerodynamic heat transfer method.

By physical reasoning, the interface scheme increases storage available to the transient external environment by representing heat capacity at skin surfaces, resulting in larger diurnal amplitudes of $Q_S$ compared with the half-layer scheme. This on

average improves performance for idealised wall and roof assemblies based on real-world construction and material properties, and where urban land surface models are under-representing heat storage magnitude. However, the interface scheme will degrade the performance of urban models if the magnitude of $Q_S$ is already well represented, which may be the case in models where thermal properties of urban surfaces are optimised to increase cyclic heat capacity, which have higher temporal and spatial resolutions, or which have an altogether different representation of urban geometry.

Other than affecting flux predictions, the interface scheme can provide structural benefits to urban land surface models. The skin temperatures of urban surfaces are used in balancing energy budgets and determining radiant and turbulent fluxes. The interface scheme calculates skin temperatures prognostically, while models using the half-layer schemes diagnose skin temperatures as an additional calculation, or assume the first layer bulk temperature is representative of the skin temperature. For aTEB, moving from a half-layer to an interface conduction scheme avoided the additional calculations required to

diagnose skin temperatures, and resulted in a 5% reduction in average runtime for offline simulations.

In conclusion, the interface conduction scheme has the potential to benefit urban land surface models simulating environmental phenomena at scales that require a high level of discretisation in time and/or space for reasons of efficiency. Examples include numerical weather prediction, where many simulations are required in short timeframes, or climate studies that require simulation over long timescales. The interface scheme also improves performance in assemblies using realistic

material thermal parameters, so may benefit large-scale studies investigating future impacts of urban design or climate mitigation measures. Results presented here are based on a single urban model with multiple configurations, and on a single observation site, so future work may extend evaluation to other sites and other urban models.

**Code/ data availability**

Source, evaluation and plotting code is included in supplementary material, available through the online git repository at:

https://bitbucket.org/matlipson/ics and further described in the README.txt. Idealised experiments (Section 3) can be run and plotted directly, however observational experiments (Section 4) require three observational dataset files that must be acquired from the data owner Andrew Coutts (andrew.coutts@monash.edu):



- alpha01.dat (used as forcing data for the urban model, includes observation quality flags)
- observation_preston.csv (flux observations to assess model performance)
- QF.txt (estimates of anthropogenic heat sources)

**Appendix A: Model parameters and material databases**

| **Table A.1**: Urban land surface model parameters (constants) | | | Units |
|---|---|---|---|
| $\delta_{(bld,veg)}$ | Area fraction: (vegetation, building) | (0.45, 0.38) | |
| $h$ | Building height | 6.4 | m |
| $h/w$ | Canyon height/width ratio | 0.42 | |
| $\alpha_{(roof,road,wall,veg)}$ | Albedo: (roof, road, wall, vegetation) | (0.175, 0.10, 0.30, 0.20) | |
| $\varepsilon_{(roof,road,wall,veg)}$ | Emissivity: (roof , road, wall, vegetation) | (0.90, 0.94, 0.85, 0.96) | |
| $Z_{0,(roof,road,wall,veg)}$ | Roughness: (roof, road, wall, vegetation) | (0.10, 0.01, 0.01, 0.25) | m |
| $d_{(1,2,3,4),road}$ | Road depth: layers (1, 2, 3, 4) | (0.01, 0.04, 0.45, 3.50) | m |
| $\lambda_{(1,2,3,4),road}$ | Road conductivity: layers (1, 2, 3, 4) | (0.7454, 0.7454, 0.2513, 0.251) | W m$^{-1}$ K$^{-1}$ |
| $C_{(1,2,3,4),road}$ | Road heat capacity: layers (1, 2, 3, 4) | (1.94E6, 1.94E6, 1.28E6, 1.28E6) | J m$^{-3}$ K$^{-1}$ |

| **Table A.2:** Urban land surface model material dataset parameters | | | | | | | | |
|---|---|---|---|---|---|---|---|---|
| | Walls | | | | Roofs | | | |
| | SITE | WRF | UZE | aTEB | SITE | WRF | UZE | aTEB |
| **Depth ($d$) [m]** | | | | | | | | |
| Layer 1 | 0.0404 | 0.05 | 0.05 | 0.05 | 0.0116 | 0.05 | 0.05 | 0.01 |
| Layer 2 | 0.054 | 0.05 | 0.05 | 0.05 | 0.05 | 0.05 | 0.10 | 0.09 |
| Layer 3 | 0.042 | 0.05 | 0.10 | 0.05 | 0.04 | 0.05 | 0.15 | 0.40 |
| Layer 4 | 0.0125 | 0.05 | 0.10 | 0.05 | 0.0125 | 0.05 | 0.20 | 0.10 |
| **Conductivity ($\lambda$) [W m$^{-1}$ K$^{-1}$]** | | | | | | | | |
| Layer 1 | 0.61 | 0.67 | 1.00 | 0.9338 | 6.53 | 0.67 | 0.40 | 1.51 |
| Layer 2 | 0.43 | 0.67 | 1.00 | 0.9338 | 0.025 | 0.67 | 0.40 | 1.51 |
| Layer 3 | 0.024 | 0.67 | 1.00 | 0.9338 | 0.23 | 0.67 | 0.40 | 0.08 |
| Layer 4 | 0.16 | 0.67 | 1.00 | 0.05 | 0.16 | 0.67 | 0.40 | 0.05 |
| **Volumetric heat capacity ($C$) [J m$^{-3}$ K$^{-1}$]** | | | | | | | | |





| | | | | | | | | |
|---|---|---|---|---|---|---|---|---|
| Layer 1 | 1.25E+06 | 1.00E+06 | 1.20E+06 | 1.55E+06 | 2.07E+06 | 1.00E+06 | 1.20E+06 | 2.11E+06 |
| Layer 2 | 1.40E+06 | 1.00E+06 | 1.20E+06 | 1.55E+06 | 7.10E+03 | 1.00E+06 | 1.20E+06 | 2.11E+06 |
| Layer 3 | 1.30E+03 | 1.00E+06 | 1.20E+06 | 1.55E+06 | 1.50E+06 | 1.00E+06 | 1.20E+06 | 0.28E+06 |
| Layer 4 | 0.67E+06 | 1.00E+06 | 1.20E+06 | 0.29E+06 | 0.67E+06 | 1.00E+06 | 1.20E+06 | 0.29E+06 |
| **Whole assembly characteristics** | | | | | | | | |
| Depth: [m] | 0.1489 | 0.2 | 0.3 | 0.2 | 0.1141 | 0.2 | 0.5 | 0.6 |
| Transmittance: [W m$^{-2}$ K$^{-1}$] | 0.495 | 3.35 | 3.333 | 0.862 | 0.444 | 3.35 | 0.80 | 0.142 |
| Cyclic heat capacity: [J m$^{-2}$ K$^{-1}$] | 104 112 | 92 002 | 143 176 | 160 349 | 27 653 | 92 002 | 80 817 | 197 755 |

### Appendix B: Aerodynamic conductance formulations

Three parameterisations of heat exchange between surfaces and turbulent air are evaluated within aTEB: the Jürges, Rowley and Harman methods. Sensible heat flux between a surface ($*$) and air is

$$Q_{H,*} = \Omega_*^{-1}(T_* - T_{air}), \tag{A1}$$

where $\Omega_*^{-1}$ is aerodynamic heat conductance $[Wm^{-2}K^{-1}]$.

### Jürges method

The Jürges method (Jürges, 1924) is implemented in aTEB as described by Kusaka et al., (2001):

$$\Omega_*^{-1} = 6.15 + 4.18U_*, \tag{A2}$$

$$\Omega_*^{-1} = 7.51U_*^{0.78}, \tag{A3}$$

where $U_*$ is the wind speed in the canyon, shared by wall and road surfaces.

### Rowley method

The Rowley method (Rowley et al., 1930) is implemented in aTEB as described by Masson (2000):

$$\Omega_*^{-1} = 11.8 + 4.2\sqrt{U_*^2 + W_*^2}, \tag{A4}$$

where $U_*$ is horizontal wind speed, and $W_*$ vertical wind speed in the canyon, shared by wall and road surfaces.

**Harman method**

The Harman method (Harman et al., 2004b) is implemented in aTEB as described by Thatcher and Hurley (2012):





$$\Omega_*^{-1} = \frac{C_{air}\,\kappa^2 U_*}{ln(0.1h/z_{0,*})(2.3+ln\,(0.1h/z_{0,*}))},\qquad\text{(A5)}$$

where $C_{air}$ is volumetric heat capacity of air, $\kappa$ is the von Kármán constant, $h$ is height of buildings, $z_{0,*}$ is the roughness length of a surface, and $U_*$ is the wind speed over a surface calculated separately for each surface and air-flow region.

**Appendix C: Variable timestep response**

In Fig. C.1 the analysis of Sect. 3.2.2 is repeated for 1, 15, 30 and 60 minute timesteps. For the half-layer scheme, the optimum combination of timestep length and layer number is achieved by increasing the number of layers to 10+ for each timestep length. For the interface scheme, an optimal number of layers can be achieved because space and time discretisation biases for storage heat flux are oppositely signed, resulting in normalised standard deviations closer to 1 (Fig. C.1(a)) and

lower mean absolute errors (Fig. C.1(b)). In higher resolution representations, the interface scheme advantage is relatively small.

**Supplement link**

To be added by publisher

**Author contribution**

M. Thatcher developed the original aTEB urban model. M. Lipson proposed and developed the interface scheme, wrote the analysis code and prepared the manuscript, with assistance from M. Hart and M. Thatcher.

**Competing interests**

The authors declare that they have no conflict of interest.

**Acknowledgements**

This study was supported by the Australian Research Council (ARC) Centre of Excellence for Climate System Science (CE110001028). Mathew Lipson was supported by an Australian Postgraduate Award. We thank Andrew Coutts for sharing original observational datasets and anthropogenic heat flux estimates. Martin Best, Sue Grimmond and Maggie Hendry for answering questions and providing additional information regarding the PILPS-Urban intercomparison analysis methods. Yannick Copin for the base Taylor diagram code. Andrew Pitman for useful feedback on the manuscript.



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



**Figures**

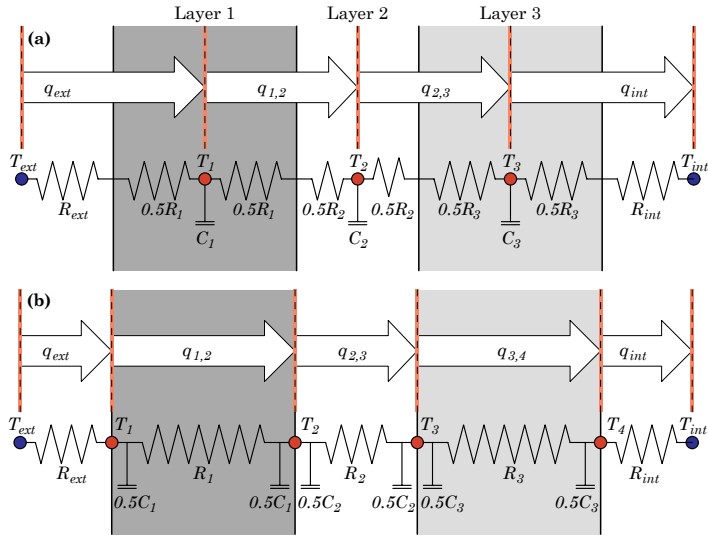

**Figure 1:** Conceptual diagram for two methods of discretising heat transfer through (three) homogenous layers. **(a)** The half-layer scheme as commonly implemented in urban land surface models, and **(b)** the proposed interface scheme, which moves temperature nodes ($T_k$) from the centre of layers to the interfaces between them, with heat capacity ($C_i$) half of each adjacent layer. Discrete paths of conduction ($q_{k,k+1}$) pass through layer resistances ($R_i$), while external and internal environment temperatures ($T_{ext}, T_{int}$) and surface thermal resistances ($R_{ext}, R_{int}$) control boundary fluxes ($q_{ext}, q_{int}$).



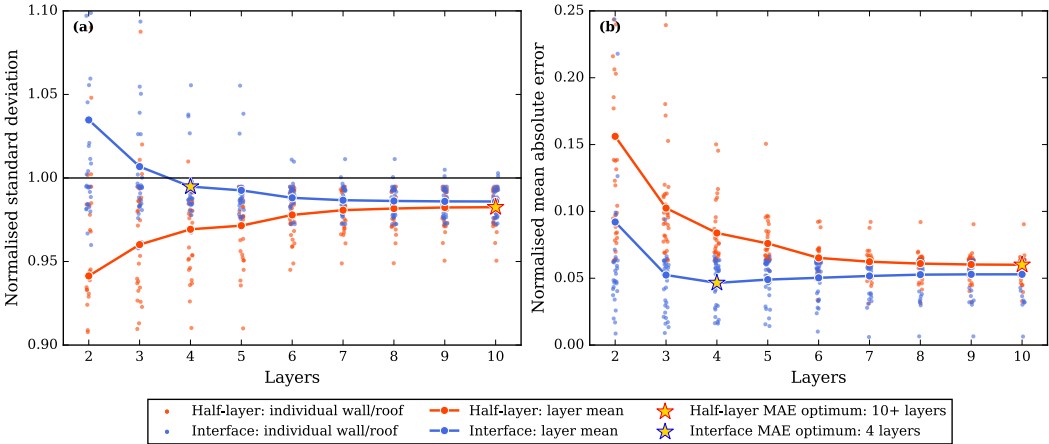

**Figure 2:** Heat storage errors of the two discrete schemes for increasing number of layers. For both (a) normalised standard deviation and (b) normalised mean absolute error, the interface scheme mean outperforms the half-layer scheme at all levels of complexity and has an optimum of four layers for 30-minute timesteps.

**Figure 3:** Heat storage errors of the two discrete schemes for increasing cyclic heat capacity (or thermal mass). For both (a) normalised standard deviation and (b) mean absolute error, the interface scheme locally weighted linear regression (LOWESS) performs better than the half-layer scheme for realistic amounts of cyclic heat capacity. Differences are more pronounced in four-layer representations.





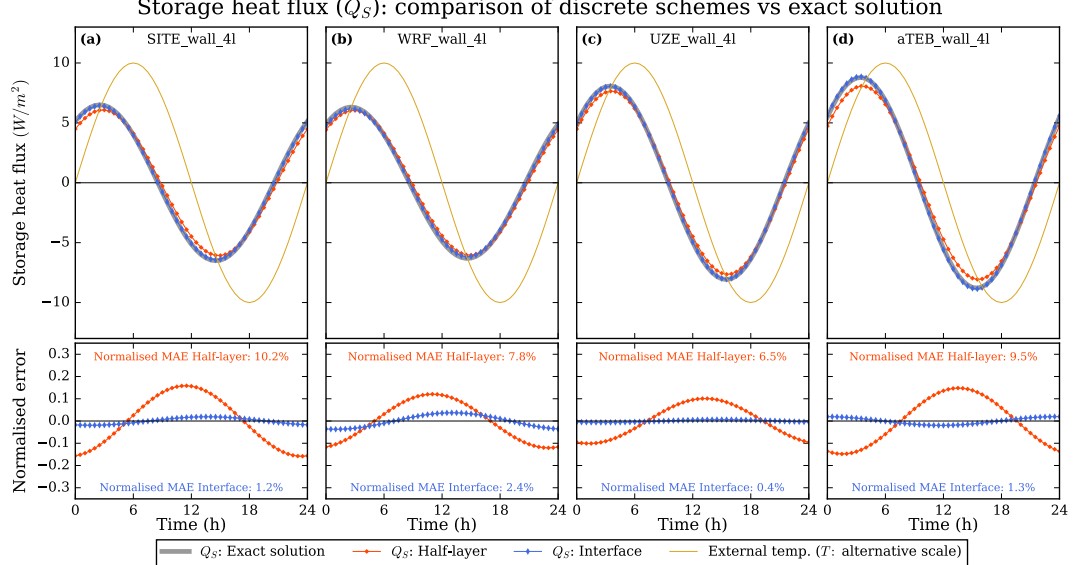

**Figure 4:** Storage heat flux ($Q_S$) response of four-layered walls to a 24h sinusoidal forcing: (a) flux density and (b) normalised error.

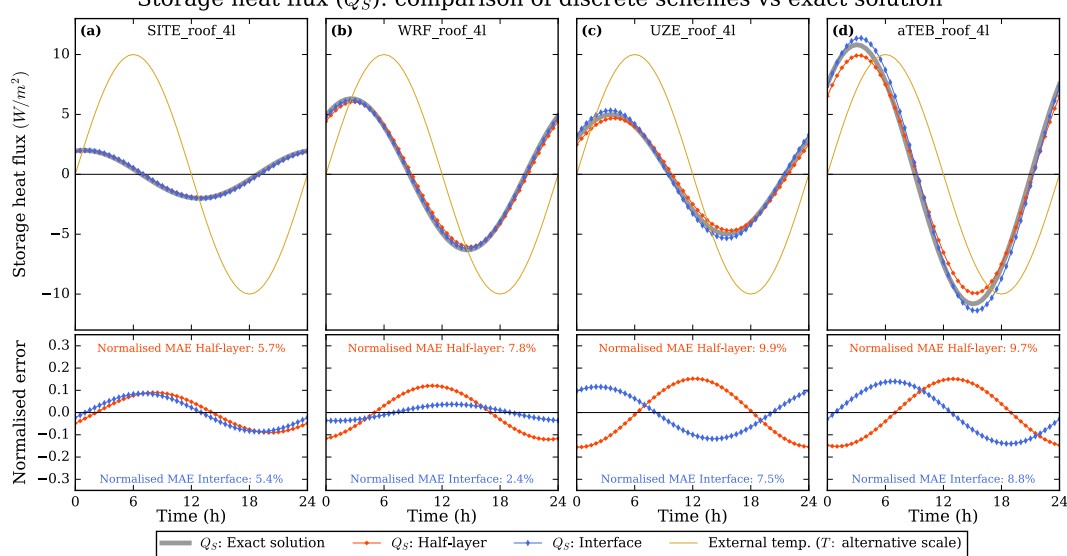

**Figure 5:** Storage heat flux ($Q_S$) response of four-layered roofs to a 24h sinusoidal forcing: (a) flux density and (b) normalised error.





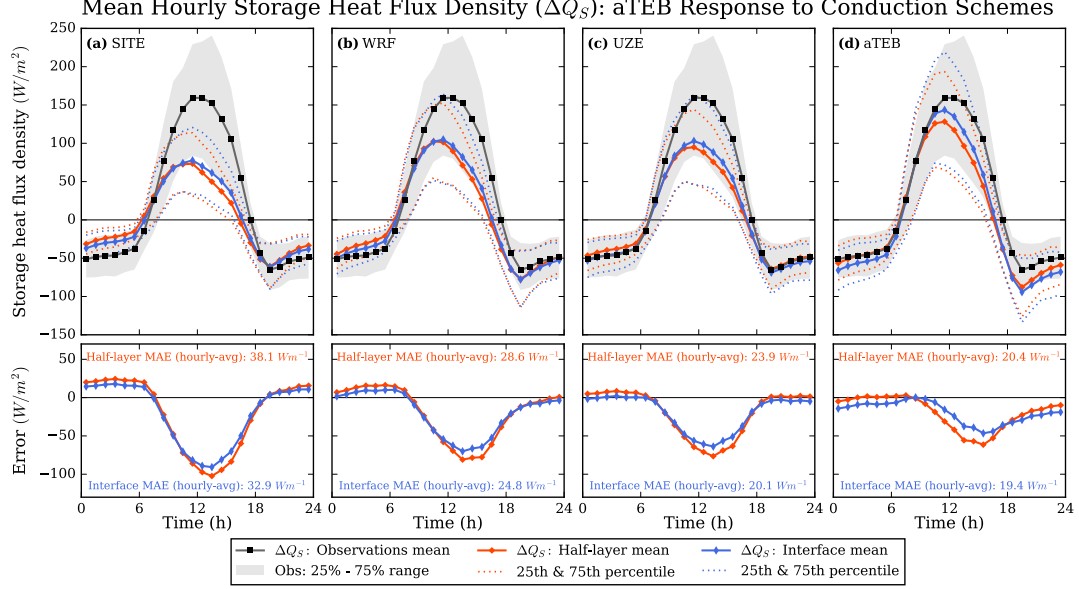

**Figure 6:** Hourly-averaged diurnal heat storage response of an urban land surface model to four wall/roof datasets over 12 months. In each case, the interface scheme reduces errors by increasing heat storage amplitude. When evaluation is limited to 3-month seasons, differences between schemes are qualitatively similar to annual results, although overall errors are higher in summer where flux magnitudes are larger (not shown). The MAE for hourly fluxes of a mean day is different to MAE of each half-hour timestep (Table 2).







**Figure 7:** A Taylor diagram of energy fluxes for each simulation with interface scheme (filled markers) and half-layer scheme (unfilled markers) connected by a line. Change toward a normalised standard deviation of 1 indicates improved variance (radial distance) and



toward the bottom axis indicates improved pattern correlation (radial angle). Where both are improved, the centred root mean square error decreases towards zero (interior arcs). Different colours indicate different material parameter datasets. Different marker shapes indicate different aerodynamic heat transfer parameterisations. Anonymous results of participants in the PILPS-urban Phase 2 intercomparison are also plotted (Grimmond et al., 2011).

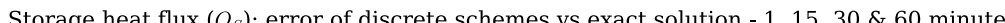

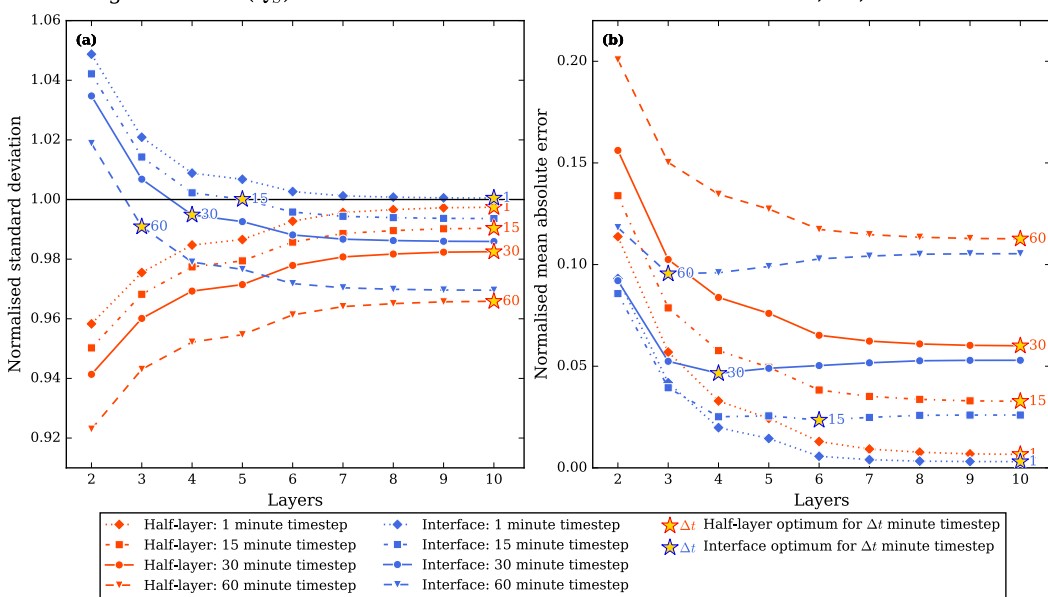

**Figure C.1:** Heat storage errors of the half-layer and discrete schemes for increasing number of layers. The 288 configuration layer means are shown for (a) normalised standard deviation and (b) normalised mean absolute errors. Optimum number of layers for each timestep is identified with a star.





**Tables**

**Table 1:** Normalised standard deviation (SD) of $Q_S$ for high-resolution (1mm deep) simulations: aTEB wall test case.

| Timestep | 1 second | 1 minute | 30 minute |
|---|---|---|---|
| Half-layer | 0.999981 | 0.999185 | 0.976463 |
| Interface | 0.999986 | 0.999190 | 0.976469 |

**Table 2**: aTEB half-hourly performance statistics for $\Delta Q_S$: difference conduction schemes (improvement in bold)

| Dataset: | SITE | | WRF | | UZE | | aTEB | |
|---|---|---|---|---|---|---|---|---|
| Scheme: | Half-layer | Interface | Half-layer | Interface | Half-layer | Interface | Half-layer | Interface |
| RMSE [W m$^2$]: | 71.60 | **66.87** | 60.80 | **57.42** | 58.13 | **54.79** | 54.03 | **51.21** |
| MAE [W m$^2$]: | 46.17 | **42.27** | 38.29 | **35.67** | 35.23 | **32.97** | 33.38 | **33.34** |
| r$^2$: | 0.66 | **0.72** | 0.74 | **0.77** | 0.80 | 0.80 | 0.78 | **0.80** |
| normalised SD: | 0.52 | **0.56** | 0.67 | **0.72** | 0.63 | **0.71** | 0.83 | **0.93** |

**Table 3:** Average performance statistics: change from half-layer to interface scheme (improvement in bold)

| Flux | RMSE [W m$^{-2}$] | MAE [W m$^{-2}$] | R$^2$ | norm. SD | norm. cRMSE |
|---|---|---|---|---|---|
| Storage: $\Delta Q_S$ | **3.27** | **2.29** | **0.017** | **0.078** | **0.033** |
| Longwave: $L_\uparrow$ | **2.38** | **1.55** | **0.010** | **0.059** | **0.062** |
| Sensible: $Q_H$ | **1.88** | **1.27** | -0.004 | **0.046** | **0.022** |
| Latent: $Q_E$ | -1.58 | -0.28 | -0.028 | **0.022** | -0.033 |

**END**