# Peer review of "Efficiently modelling urban heat storage: an interface conduction scheme in an urban land surface model (aTEB v2.0)"

_Geoscientific Model Development, 2016_

## Short Comment (SC1) · 24 Oct 2016

Dear authors,

In my role as Executive editor of GMD, I would like to bring to your attention our Editorial version 1.1:

http://www.geosci-model-dev.net/8/3487/2015/gmd-8-3487-2015.html

This highlights some requirements of papers published in GMD, which is also available on the GMD website in the 'Manuscript Types' section:

http://www.geoscientific-model-development.net/submission/manuscript_types.html

In particular, please note that for your paper, the following requirement has not been met in the Discussions paper:

- "The main paper must give the model name and version number (or other unique identifier) in the title."

Please add a version number for aTEB in the title upon your revised submission to GMD.

Yours,

Astrid Kerkweg

---

## Referee Comment (RC1) · Anonymous Referee #1 · 13 Dec 2016

This paper presents a new heat conduction scheme which can be implemented in the urban land surface models and compares it with a well-established and widely used scheme in current land surface models. The study is interesting and in the scope of the Geoscientific Model Development. The author(s) made a reasonable effort and the research was carefully conducted. I think that this paper could be published promptly because the new proposed scheme and the discussions in the paper are helpful for the developers and users of the urban land surface models. However, there are some flaws in the manuscript, which I think that the authors should consider to revise before the manuscript is finally accepted.

Major: 1. One of my major concerns is about the structure of the manuscript. I feel a

little confused when I read through the manuscript and had to go back and forth for a few times. I think it is clearer that if the authors can restructure section 2, into which part of section 3 and 4 can be moved. The new section 2 serves as a Methods section, where the authors introduce the two conduction schemes, the aTEB model, modeling setup, idealized method, as well as the data used. Then section 3 and section 4 serve as results sections to discuss about the idealized results and observational results.

2. The authors use two methods to assess the performance of the two schemes: idealized environment and observational datasets. When using observational dataset, the improvement in Qs and other fluxes when changing from half-layer scheme to interface scheme are rather small ($< 5$ W m-2). I wonder whether the magnitudes of these improvements are statistically significant? Could the authors provide any statistics to prove that?

Minor: 1. Page 2 Line 3: "the alternate method" -> "the alternative method" 2. Captions in Figure 4 and Figure 5: (a) flux density and (b) normalized error. I suppose the authors mean: (top panel) flux density and (bottom panel) normalized error

---

## Referee Comment (RC2) · J. Kala (Referee) · 21 Dec 2016

This paper is concerned with the storage heat flux in areas with large thermal mass and multiple surface layers. It provides a really nice, through and useful analysis of the impact of where temperature nodes are in a layered system. My comments are very minor and easily addressed.

Comments

1) Notation is confusing for the storage heat flux as two sets of symbols are used for what should be the same (as the model is being evaluated against it- but is referred to differently between the model and observations
   a. P1. L20 should be $\Delta Q_S$ (that notation throughout)
   b. P2L23 - $\Delta$ changing notation – Infer that the $\Delta$ is related to the residual whereas in fact for both it should be the net change in all components of the system e.g. trees, air, as well as the built materials. The observational method of a residual difference is not related to $\Delta$.  The notation is confusing relative to the net change in storage – the authors could create a subscript to distinguish the two but refer to it with the same notation or refer to observed as RES (residual) as commonly done

2) P2. L2 – there are multiple methods - so it is an additional method
3) P3, l5 – cite the original Grimmond et al. 1991
4) P3 l12 – references should be in chronological order (throughout)
5) P3 L29 – Homogenous – reference examples
6) P4 l3 – Jackson et al. (**date**)
7) P6 l1&2 – int – for the second $R_{int}$
8) P6 l23 – for the respective time step?
9) P8l25 l assemblies or assemblages?
10) P8 l24 – give accessible reference location
11) P9 line 2 – are representative of? The central tendency of … later in sentence 'represent' or 'are for the …'
12) P9/L18 – values have been multiplied by 100 %  to be made a percent
13) P9 l 19 – reword sentence
14) P10 – heading observational methods – some aspects have already been introduced, why this heading here? And why is the description of the model under observations?
15) P11 line 5 – net all-wave radiation
16) P11 line 25 – notation now includes $\Delta$
17) P12 line 16 – define centred RMSE
18) P14 top paragraph – consider work Salamanca, F., **E**.S. Krayenhoff, A. Martilli. 2009: On the derivation of material thermal properties representative of heterogeneous urban neighbourhoods, *J. Appl. Meteorol. Climatol.* **48**, 1725-1732
19) P14 l21 enhance rather than benefit?
20) Figure and Table caption in general need to be standalone   so the material can be understood
21) Appendix A –Tables  A1 and A2 give references or basis for data  sources here or indicate where these are given in the text. SITE, WRF uZE aTEB – need to be defined to
22) Figures 2 onwards –  - units should have space between the dimensions. The extra headers should be removed (difference between talk and paper presentation)
23) Figure 4 and 5 – give more details of types of conditions used
24) Figure 6 – Caption is not complete enough to be stand alone. Need correct spacing on units W m$^{-2}$ - fix units within plots
25) Figure 7 – Label Y axis

---

## Author Comment (AC1) · 24 Jan 2017

Thank you for your comment. The manuscript title will be revised from:

Efficiently modelling urban heat storage: an interface conduction scheme in the aTEB urban land surface model

to:

Efficiently modelling urban heat storage: an interface conduction scheme in an urban land surface model (aTEB v2.0)

---

## Author Comment (AC2) · 27 Jan 2017

This document is the authors response to the comments of Referee 1 (RC1) for the manuscript: http://www.geosci-model-dev-discuss.net/gmd-2016-240/

Page and line references (P# L#) are listed for original manuscript (in red with referee comments) and for the revised manuscript (in black with response).

This paper presents a new heat conduction scheme which can be implemented in the urban land surface models and compares it with a well-established and widely used scheme in current land surface models. The study is interesting and in the scope of

the Geoscientific Model Development. The author(s) made a reasonable effort and the research was carefully conducted. I think that this paper could be published promptly because the new proposed scheme and the discussions in the paper are helpful for the developers and users of the urban land surface models. However, there are some flaws in the manuscript, which I think that the authors should consider to revise before the manuscript is finally accepted.

Major: 1. One of my major concerns is about the structure of the manuscript. I feel a little confused when I read through the manuscript and had to go back and forth for a few times. I think it is clearer that if the authors can restructure section 2, into which part of section 3 and 4 can be moved. The new section 2 serves as a Methods section, where the authors introduce the two conduction schemes, the aTEB model, modeling setup, idealized method, as well as the data used. Then section 3 and section 4 serve as results sections to discuss about the idealized results and observational results.

We thank Referee 1 for their advice regarding the structure of the manuscript, which we have adopted. We feel these changes will help make the manuscript easier to read and understand. The parts of Section 3 and 4 which described methods and model description has been moved to Section 2. The remaining parts of Section 3 and 4 have been combined into a results Section 3. Discussion and conclusion Section 5 has become Section 4 and some other section titles have been revised. A snapshot of the structural changes are attached at the end of this document.

[Major] 2. The authors use two methods to assess the performance of the two schemes: idealized environment and observational datasets. When using observational dataset, the improvement in Qs and other fluxes when changing from half-layer scheme to interface scheme are rather small (< 5 W m-2). I wonder whether the magnitudes of these improvements are statistically significant? Could the authors provide any statistics to prove that?
As suggested, we have extended the study to test the statistical significance of mean improvements for all fluxes (Table 3). We undertake a paired, two-sided T-test for the null hypotheses that two dependent samples have the same mean values (i.e. we test the significance of the improvement of the interface scheme). In each case, the 95% confidence interval is reached. We note the significance in the text (last line of Taylor diagram section, previously P13 L6), and in the body and caption of Table 3.

Minor: 1. Page 2 Line 3: "the alternate method" -> "the alternative method"

P2 L2: Sentence has been reworded and "the alternate method" removed.

[Minor]2. Captions in Figure 4 and Figure 5: (a) flux density and (b) normalized error. I suppose the authors mean: (top panel) flux density and (bottom panel) normalized error

Yes, that was the intention, it has been revised.

Again, we thank the reviewer for these comments.

———————————————————

▼ 1 Introduction
    1.1 Measuring storage heat flux in cities
    1.2 Simulating storage heat flux in cities
    1.3 Material thermal parameters
▼ 2 Description of conduction representations
    2.1 Half-layer scheme
    2.2 Interface scheme
    2.3 Exact solution
▼ 3 Idealised evaluation
    3.1 Idealised method
    ▼
        3.2.1 Idealised results: high-resolution simulation
        3.2.2 Idealised results: realistic material parameters
        3.2.3 Idealised results: typical material parameters
▼ 4 Observational evaluation
    ▼ 4.1 Observational methods
        4.1.1 Description of urban model: aTEB
        4.1.2 Observational data
    ▼ 4.2 Observational results
        4.2.1 Observational results: heat storage
        4.2.1 Observational results: other fluxes
5 Discussion and Conclusion

▼ 1 Introduction
    1.1 Measuring storage heat flux in cities
    1.2 Simulating storage heat flux in cities
    1.3 Material thermal parameters
▼ 2 Methods
    ▼ 2.1 Conduction representations
        2.1.1 Half-layer scheme
        2.1.2 Interface scheme
        2.1.3 Exact solution
    2.2 Idealised evaluation methods
    ▼ 2.3 Urban model evaluation methods
        2.3.1 Description of urban model: aTEB
        2.3.2 Observational data
▼ 3 Results
    ▼ 3.1 Idealised evaluation results
        3.1.1 High-resolution simulation
        3.1.2 Realistic material parameters
        3.1.2 Typically modelled parameters
    ▼ 3.2 Urban model results
        3.2.1 Impact on heat storage
        3.2.2 Impact on other fluxes
4 Discussion and Conclusion

**Fig. 1.** The old (left) and new (right) outlines.

---

## Author Comment (AC3) · 27 Jan 2017

**Efficiently modelling urban heat storage: an interface conduction scheme in an urban land surface model (aTEB v2.0)**

Mathew J. Lipson[1], Melissa A. Hart[1], Marcus Thatcher[2]

[1]Climate Change Research Centre, UNSW and ARC Centre of Excellence for Climate System Science
[2]CSIRO Marine and Atmospheric Research, Aspendale, Australia

*Correspondence to*: M. Lipson (m.lipson@unsw.edu.au)

**Summary**

This document is the authors response to the Referee reviews and discussion of the manuscript:
**http://www.geosci-model-dev-discuss.net/gmd-2016-240/**

Page and line references (P# L#) are listed for original manuscript (in red with referee comments) and for the revised manuscript (in black with author response).

**RC2: Anonymous Referee #2**

*This paper is concerned with the storage heat flux in areas with large thermal mass and multiple surface layers. It provides a really nice, through and useful analysis of the impact of where temperature nodes are in a layered system. My comments are very minor and easily addressed.*

We thank Referee # 2 for these positive comments. The thoroughness of their review has improved the paper.

*1) Notation is confusing for the storage heat flux as two sets of symbols are used for what should be the same (as the model is being evaluated against it- but is referred to differently between the model and observations*

> *a. P1 L20 should be ΔQS (that notation throughout)*

P1 L20: We agree and have changed notation to $\Delta Q_S$ throughout.

> *b. P2 L23 - Δ changing notation – Infer that the Δ is related to the residual whereas in fact for both it should be the net change in all components of the system e.g. trees, air, as well as the built materials. The observational method of a residual difference is not related to Δ. The notation is confusing relative to the net change in storage – the authors could create a subscript to distinguish the two but refer to it with the same notation or refer to observed as RES (residual) as commonly done*

Revised P2 L24: In the original manuscript, we made a distinction between modelled net storage flux ($Q_S$) and observed net storage flux calculated as a residual ($\Delta Q_S$). As noted the notation differs from established forms which has led to confusion, and on reflection is not necessary. We now refer only to $\Delta Q_S$, meaning net storage heat flux density. Where a distinction is important, we note whether it is observed or modelled.

*2) P2 L2 – there are multiple methods - so it is an additional method*

Revised P2 L2: We have amended this sentence to make clear we are comparing a new alternative method with a well-established method.

*3) P3 L5 – cite the original Grimmond et al. 1991*

P3 L5: Done

*4) P3 L12 – references should be in chronological order (throughout)*

P3 L11: Done

*5) P3 L29 – Homogenous – reference examples*

Revised P3 L29: We have reworded sentence to make clearer the distinction between homogenous and composite materials, and updated other instances of "layered" materials to "composite" materials throughout. We have also changed '*some models use…*' in this line to '*modellers sometimes describe...*' to make a distinction that it is often a modellers choice to represent materials as homogenous (as opposed to being hardwired in a model). Lastly, we have moved the database descriptions (CLMU, SITE etc) to an Appendix for clearer flow of the article.

*6) P4 L3 – Jackson et al. (date)*

P14 L19: Done

*7) P6 L1&2 – int – for the second $R_{int}$*

P5 L9: Done

*8) P6 L23 – for the respective time step?*

P6 L23: Deleted lines 20-23 as they are not necessary and are confusing.

*9) P8 L25 I assemblies or assemblages?*

P10 L2: Assemblies, as an ordered group of components that work together, rather than an unordered assemblage.

*10) P8 L24 – give accessible reference location*

P10 L2: Done, referenced Fig. D.1

*11) P9 line 2 – are representative of? The central tendency of ... later in sentence 'represent' or 'are for the ...'*

P10 L9: Done, replaced "are representative of" with "represent".

*12) P9/L18 – values have been multiplied by 100 % to be made a percent*

P10 L24: We are unclear as to what change was required as the values on p9 are in the a 0-100% range.

*13) P9 L19 – reword sentence*

P10 L25: Done.

*14) P10 – heading observational methods – some aspects have already been introduced, why this heading here? And why is the description of the model under observations?*

Manuscript has been restructured per Referee #1 comments (see RC1 response). This section moved to P8 L16.

*15) P11 L5 – net all-wave radiation*

P8 L18: Done.

*16) P11 L25 – notation now includes Δ*

P11 L5: All $Q_S$ now referred to as $\Delta Q_S$, see response to 1b).

*17) P12 L16 – define centred RMSE*

P11 L29: Referenced Taylor (2001).

*18) P14 top paragraph – consider work Salamanca, F., E.S. Krayenhoff, A. Martilli. 2009: On the derivation of material thermal properties representative of heterogeneous urban neighbourhoods, J. Appl. Meteorol. Climatol. 48, 1725-1732*

P13 L15: The authors are aware of the interesting work by Salamanca et. al (2009), which considers various methods to aggregate material thermal properties (depth, conductivity and heat capacity) to give an overall homogenous material that is meant to be representative of patches of different urban materials. However, the current manuscript uses 'typically modelled' material thermal parameters sourced from previously published studies, and avoids creating new sets of thermal parameters. Therefore, the method described in Salamanca et. al (2009) was not considered relevant in this study.

*19) P14 L21 enhance rather than benefit?*

P13 L29: We feel in this case benefit is more appropriate, as enhance may suggests additional features, which is not the case.

*20) Figure and Table caption in general need to be standalone so the material can be understood*

Done, details below.

*21) Appendix A –Tables A1 and A2 give references or basis for data sources here or indicate where these are given in the text. SITE, WRF uZE aTEB – need to be defined to*

Done.

*22) Figures 2 onwards – - units should have space between the dimensions. The extra headers should be removed (difference between talk and paper presentation)*

Done

*23) Figure 4 and 5 – give more details of types of conditions used*

Done

*24) Figure 6 – Caption is not complete enough to be stand alone. Need correct spacing on units W m$^{-2}$ – - fix units within plots*

Done

*25) Figure 7 – Label Y axis*

Done

We thank the reviewer for these valuable comments.

---

## Author Comment (AC4) · 27 Jan 2017

**Efficiently modelling urban heat storage: an interface conduction scheme in an urban land surface model (aTEB v2.0)**

Mathew J. Lipson[1], Melissa A. Hart[1], Marcus Thatcher[2]

[1]Climate Change Research Centre, UNSW and ARC Centre of Excellence for Climate System Science
[2]CSIRO Marine and Atmospheric Research, Aspendale, Australia

*Correspondence to*: M. Lipson (m.lipson@unsw.edu.au)

**Summary**
Part A includes a summary of changes not listed in response to referee comments RC1 or RC2. Part B includes the full manuscript with changes tracked.

Analysis and plotting code has also been updated to accommodate referee comments. The original and revised code and figures can be compared here: https://bitbucket.org/matlipson/ics/commits/all

**Part A – Other changes**

**1) P1 L15**

P1 L14: Added 'of common'

**2) P1 L19-25**

P1 L19-25: Reworded first sentence in Introduction for clarity. Moved second sentence on land surface models down to P1L26 to locate with introduction of model intercomparison. Deleted some unnecessary words in third sentence for simplicity.

**3) P2 L11**

P2 L12: Where appropriate throughout replaced "surfaces" with "materials" to make clearer distinction with the skin surface of urban materials.

**4) P2 L15**

P2 L16: Added net ($\Delta$) to advection flux $Q_A$ for consistency with Referee #2 comment on net storage flux $\Delta Q_S$.

**5) P3 L6-9**

P3 L6-9: Reworded two sentences for clarity. Added reference for PILPS-Urban. Made clearer reference to Best and Grimmond 2014b finding on methods for calculating net heat storage.

**7) P7 L11 (end of 2.3 Exact Solutions)**

P6 L20: Added the following paragraph to better describe the effective heat capacity presented in Fig. 3 and discussed in the paper:

*Periodic areal heat capacity (ISO 13786:2007) is a useful measure of a composite materials ability to store heat over a sinusoidal cycle. It is a better measure than overall heat capacity or surface thermal admittance as it accounts for the periodic penetration depth of each material layer (for thick composites) as well as heat lost through transmittance (for thin composites). It can be calculated exactly as:*

$$\kappa = \frac{P}{2\pi} \left| \frac{H_{22}-1}{H_{12}} \right|, \tag{20}$$

*with units J m$^{-2}$ K$^{-1}$.*

**8) P15 L1**

P14 L10: Added link to git repository in Code/ data availability.

**11) Appendix A&B**

Split into two appendices, A for material database information, B for model parameters. Moved material database description from body of manuscript to Appendix A (previously in Section 1.3). Updated other Appendix names and references. Renamed Table A2 to Table A1. Deleted 'transmittance' information from Table as not discussed in the manuscript.

**9) Appendix C**

Added wind speed information to formulae A2 & A3.

**10) Figure 7**

Made changes to improve clarity of Taylor diagram including adding y-axis label (per RC2 comment 25), adding label for cRMSE and making panel titles bold to stand out.

**Part C – Revised Manuscript Changes**

All changes from the originally submitted manuscript, with changes marked:

-
- Added items are blue and underlined
- Moved items are green, either  (moved from) or underlined (moved to).

[revised manuscript text omitted]